# Blockade on Lin28a Prevents Cognitive Impairment and Disruption of the Blood-Brain Barrier Induced by Chronic Cerebral Hypoperfusion

**DOI:** 10.3390/biomedicines10040852

**Published:** 2022-04-05

**Authors:** Jae-Min Lee, Joo Hee Lee, Youn-Jung Kim

**Affiliations:** 1College of Nursing Science, Kyung Hee University, Seoul 02447, Korea; sunjaesa@hanmail.net; 2Korea Armed Forces Nursing Academy, Daejeon 34059, Korea; lovejjoo2@naver.com

**Keywords:** Lin28a, vascular dementia, cognitive impairment, blood-brain barrier, chronic cerebral hypoperfusion

## Abstract

Lin28a is an RNA-binding protein involved in the translation and regulation of multiple mRNAs. Lin28a is overexpressed in animal models of brain injury. Similarly, our preliminary study found increased Lin28a expression levels in the animal models four to seven days after chronic cerebral hypoperfusion. Therefore, this current study aimed to evaluate the effects of modulating Lin28a on cognition and brain functions. Vascular dementia (VaD) was induced in 12-week-old male Wistar rats using permanent bilateral common carotid artery occlusion (BCCAO), and these rats were treated with Lin28a siRNA on the fourth and seventh day after BCCAO. From the 42nd day after BCCAO, cognitive behavioral experiments were performed for two weeks. VaD induced by BCCAO resulted in cognitive impairment and microglial activation. Lin28a expression was upregulated after BCCAO. Lin28a siRNA treatment alleviated cognitive impairment and overexpression of GFAP and Iba-1 in the brain. Furthermore, the treatment ameliorated the VaD-induced damage to the blood-brain barrier (BBB) components, including PECAM-1, PDGFRβ, occludin, claudin-9, and ZO-1. CCR6 activation after VaD, associated with BBB disruption, was diminished by treatment with Lin28a siRNA. The treatment inhibited VaD-induced microglial activity and alleviated BBB damage. Thus, blocking Lin28a may alleviate cognitive impairment caused by VaD.

## 1. Introduction

Lin-28 is a highly conserved RNA-binding protein which regulates the translation of several proteins and the biosynthesis of specific miRNAs [1]. It mediates various cellular processes, including glia formation, embryogenesis, myogenesis, glucose metabolism, cancer progression, germ cell development, and cellular differentiation [2,3]. Lin28a is expressed in embryonic stem cells and early stage embryos, after which its level rapidly decreases—it is rarely expressed in adulthood [4]. The increase in Lin-28 expression level is associated with tumorigenesis and tumor growth, and it directly contributes to the regulation and maintenance of cancer stem cells [5]. Lin-28 is transiently upregulated after intracerebral hemorrhage, and overexpression of Lin28a causes astrocyte hyperplasia [6].

Vascular dementia (VaD) is the second most common type of dementia after Alzheimer’s disease and causes brain lesions due to reduced blood flow to the brain [7]. Chronic cerebral hypoperfusion causes vascular dementia, owing to the damage of microvessels in the brain [8]. Chronic cerebral hypoperfusion-induced VaD results in neuronal death, neuroinflammation, oxidative stress, excitotoxicity, neurogenesis, gliogenesis, and angiogenesis [9]. Chronic cerebral hypoperfusion, which causes VaD, can be induced by permanent bilateral common carotid artery occlusion (BCCAO), which results in the activation of microglia and reactive astrocytes due to reduced cerebral blood flow [10]. Following BCCAO, microglia rapidly migrate to the lesion site and produce inflammatory cytokines and cytotoxic substances. Microglial activation promotes neuronal apoptosis and causes brain damage. The activation of astrocytes due to vascular dementia increases the expression levels of metalloproteinases, causing blood-brain barrier (BBB) damage [11,12].

The foundation of BBB is formed by the cerebral microvascular endothelium, which, together with the basal lamina, astrocytes, microglia, pericytes, tight junctions, neurons, and extracellular matrix, constitute a neurovascular unit essential for the maintenance of homeostasis and functioning of the central nervous system [13]. VaD disrupts the BBB and damages microvessels in the brain. BBB disruption is an important pathophysiological process in ischemic stroke that leads to cerebral vasogenic edema, leukocyte infiltration, and leakage of toxic molecules into the brain [14]. After chronic cerebral hypoperfusion, the attenuation of tight junction proteins, including ZO-1, occludin, and claudin, increases BBB permeability and induces cerebral vasogenic edema. Our previous study demonstrated that damage to the BBB is caused by BCCAO-induced cognitive impairment [15]. Microglial activation causing BBB disruption releases cytokines to activate reactive astrocytes, thereby increasing apoptosis and neuroinflammation [16,17].

The rise in neuroinflammation increases the levels of interleukin 1 beta (IL-1β), tumor necrosis factor-alpha (TNF-α), and CC chemokine receptor 6 (CCR6). CCR6 has been implicated in several diseases, including spinal cord injury, traumatic brain injury, and stroke, and is upregulated by cytokines during neurodegeneration [18,19]. CCR6 plays a role in neuroinflammation induced by ischemic stroke. CCR6-knockout mice are protected against brain damage from stroke [20].

Overexpression of Lin28a in the brain is related to the activation of microglia and reactive astrocytes after chronic cerebral hypoperfusion. It is important to suppress the overactivation of microglia and reactive astrocytes caused by VaD. This study aimed to evaluate the effects of blocking Lin28a, which may reduce cognitive impairment and BBB disruption caused by VaD.

## 2. Materials and Methods

Eight-week-old male Wistar rats weighing 180 ± 20 g were purchased from Orient Bio (Seongnam, Gyeonggi-do, Korea). The animals were housed in a place where water and food were freely consumed, the environment was regulated (22 ± 2 °C, humidity of 50%), and day and night cycle (12 h light/12 h dark) was controlled. This study was conducted in accordance with the National Institutes of Health Guide for the Care and Use of Laboratory Animals and was approved by the Institutional Animal Care and Use Committee (IACUC) of Kyung Hee University (KHSASP-19-080).

### 2.1. Bilateral Common Carotid Arteries Occlusion (BCCAO) Procedure

Prior to surgery, the rats were anesthetized with 2% isoflurane in 70% N_2_O, balanced with O_2_. BCCAO was performed, avoiding damage to the vagus nerve and the surrounding tissues. Both carotid arteries were exposed by a ventral midline incision and double-ligated with 3-0 silk (Black silk; Ailee, Co., Ltd., Seoul, Korea) immediately below the carotid bifurcation. Sham animals underwent the same procedure, but without vessel ligation.

### 2.2. Experimental Design

Male Wistar rats (aged 12 weeks) were randomly divided into three groups (*n* = eight in each group): sham, VaD, and VaD + siRNA Lin28a. After BCCAO, the rats were randomly divided into VaD and VaD + siRNA Lin28a groups. The siRNA was designed to target the rat Lin28a sequence 5′-TGGACGTCTTTGTGCACCAGAGC-3′ (siRNA, accession number NM_001109269.1; Bioneer, Co., Ltd., Daejeon, Korea). After making a hole with a stereotaxic attachment drill, 3 μL (500 pmol/μL) of siRNA was injected into the pial surface (−0.8 mm anterior to the bregma, 1.4 mm lateral to the midline, and −3.6 mm beneath the dura) with a Hamilton syringe for 5 min. After the injection, the needle was left for an additional 5 min.

### 2.3. Behavior Tests

#### 2.3.1. Passive Avoidance Test

Passive avoidance test is used to evaluate the memory function based on negative reinforcement [21]. The passive avoidance device (GEMINI™ Avoidance System, San Diego Instruments, San Diego, CA, USA) has two chambers and the floor is composed of a wire mesh. The rats were kept in a dark chamber for 1 min and then moved to the opposite chamber where the light was turned on. The rats received an electric shock (0.5 mA) for 2 s when moving from the light chamber to the dark chamber. After 24 h, the latency time taken to enter the dark chamber from the light chamber when the light was turned on was recorded, with a maximum of 300 s recorded.

#### 2.3.2. Radial 8-arm Maze Test

Radial eight-arm maze test measures spatial learning and working memory [22]. After six weeks of BCCAO, the number of correct and incorrect choices were analyzed using the radial 8-arm maze test. A small container filled with 20 μL of water was placed at the end of each arm in the radial 8-arm maze device. The rats were deprived of water for 48 h and then allowed to explore and drink water for 8 min. After 24 h, the test section recorded the correct number of times it reached each arm and sought water for over 8 min. Re-entering the previously visited arm was counted as an error and the number of times all water in each arm was found was recorded as an error.

### 2.4. Tissue Preparation

Rats were anesthetized by inhalation with ether, perfused with 0.01 M phosphate buffered saline (PBS), and the brains were removed. For western blot experiments, the samples were stored at −70 °C until use. For immunohistochemistry, after perfusion with PBS, followed by additional perfusion with 4% paraformaldehyde (PFA), the brain was removed and fixed overnight in 4% PFA, followed by fixation in 30% sucrose solution. Subsequently, they were sliced into 40 μm coronal sections on a cryostat (microtome, CM3050S, Leica, Nussloch, Germany).

### 2.5. Immunohistochemistry and Immunofluorescence

Immunohistochemistry for the visualization of Lin28a, glial fibrillary acidic protein (GFAP), Iba-1 expression was performed. For the blocking process, 10% normal goat and donkey serum and 1% BSA + 0.3% Triton X-100 in PBS were blocked for 90 min at RT. In brief, sections were incubated overnight at 4 °C with Lin28a (1:1000, Abcam, Cambridge, UK), GFAP (1:1000, Abcam, Cambridge, UK), Iba-1 (1:1000, Abcam, Cambridge, UK). Sections were then incubated with the biotinylated rat or mouse secondary antibody (1:200, Vector Laboratories, Berlingame, CA, USA) with 0.3% Triton X-100 in PBS for 2 h at RT and subsequently incubated with avidin-biotin complex solution (Vector Elite ABC kit®; Vector Laboratories, Berlingame, CA, USA) for 90 min at RT. Finally, sections were stained with 3,3′-diaminobenzidine (DAB). Sections were mounted on gelatin-coated slides and air-dried for 24 h at RT. The coverslips were mounted using Permount^®^ (Vector Laboratories, Berlingame, CA, USA).

Immunofluorescence was performed to visualize GFAP, Lin28a, occludin, PDGFRβ, and rat endothelial cells antigen-1 (RECA-1) expression After blocking (10% normal goat and donkey serum and 2% BSA + 0.3% Triton X-100 in PBS) for 90 min, sections were incubated overnight at 4 °C with a mixture of two of the following primary antibodies: RECA-1 (1:1000, Abcam, Cambridge, UK), occludin (1:1000, LifeSpan BioSciences, Seattle, WA, USA), and PDGFRβ (1:500, Abcam, Cambridge, UK). Sections were incubated with Alexa Fluor® 488-conjugated affinipure donkey anti-mouse IgG mixture and Alexa Fluor® 594-conjugated affinipure goat anti-rabbit IgG (1:2000, Molecular Probes, Eugene, OR, USA) with and 0.3% Triton X-100 in PBS for 2 h at RT. For nucleus staining, nuclei were visualized with 4′,6-diamidino-2-phenylindole (DAPI). Stained slides were photographed using a confocal microscope (LSM 700, Zeiss, Oberkochen, Germany), and Image-Pro^®^ Plus software (Media Cybernetics, Silver Spring, MD, USA) was used for quantitative analysis.

### 2.6. Western Blot

Proteins were extracted from rat brain tissues by radioimmunoprecipitation assay (RIPA) buffer (ThermoFisher Scientific, Waltham, MA, USA). Protein (20 µg) was fractionated by 8–12% SDS-PAGE gel and transferred to a PVDF. After incubation with 5% skim milk in TBST (10 mM Tris, pH 7.6, 150 mM NaCl, 0.1% Tween 20) for 1 h at RT, PECAM-1 (1:1200, LifeSpan BioSciences, Seattle, WA, USA), PDGFRβ (1:500, Abcam, Cambridge, UK), occludin (1:1000, LifeSpan BioSciences, Seattle, WA, USA), claudin-9 (1:1500, Proteintech, Chicago, IL, USA), ZO-1 (1:100, LifeSpan BioSciences, Seattle, WA, USA), e-NOS (1:1000, Enzo Life Sciences, Farmingdale, NY, USA), Laminin (1:1000, ab11575, Abcam, Cambridge, UK), and β-actin (1:100,000, Santa Cruz, CA, USA) were incubated overnight at 4 °C. Membranes were incubated with a 1:3000 dilution of horseradish peroxidase-conjugated anti-rabbit or mouse or goat secondary antibodies for 90 min. Blots were developed with enhanced chemiluminescence (Clarity™ Western ECL Substrate, Bio-Rad, Hercules, CA, USA). The expression level of the protein band was quantified using Image J software (National Institutes of Health, Bethesda, MD, USA).

### 2.7. Statistical Analysis

Statistical analysis of all data was performed using SPSS version 26.0 (IBM SPSS, IL, USA). All data were presented as means ± standard error of the mean (S.E.M). All data, including behavioral experimental results, were analyzed using One-way ANOVA followed by a Tukey post hoc test. Differences between groups were considered significant at *p* < 0.05.

## 3. Results

### 3.1. Lin28a Expression Was Increased in the Hippocampal Dentate Gyrus after VaD

Normally, Lin28a is found during early embryonic and fetal development in several organs but is gradually reduced and no longer expressed in adulthood. Changes of Lin28a expression in the hippocampal dentate gyrus on 4, 7, 14, 28, and 42 days after BCCAO are shown in Figure 1. At 4, 7, 14, and 28 days after BCCAO, the expression of Lin28a was significantly increased compared to the sham group (F = 15.234, *p* < 0.001). At 42 days after BCCAO, there was no change in Lin28a expression. In particular, the greatest overexpression of Lin28a was observed on day seven after BCCAO. Our findings show that Lin28a overexpression is increased from early to 28 days after BCCAO.

### 3.2. Iba-1 and GFAP Expression Was Increased in the Hippocampal Dentate Gyrus after VaD

Immunostaining for Iba-1 and GFAP in the hippocampal dentate gyrus was performed to investigate the activation of microglia and reactive astrocytes. As shown in Figure 2, At 4, 7, 14, 28, and 42 days after BCCAO, the expression of Iba-1 (F = 11.686, *p* < 0.001) and GFAP (F = 28.782, *p* < 0.001) was significantly increased compared to the sham group. The VaD group showed thicker and larger morphological changes in microglia and astrocytes in the hippocampal dentate gyrus.

### 3.3. Blockade of Lin28a Ameliorates Loss of Tight Junction Proteins after VaD

Microvascular damage and BBB damage are prominent pathological features of chronic cerebral hypoperfusion. As shown in Figure 3A, occludin and PDGFRβ expression in the VaD group were decreased compared to the sham group. In addition, the morphology of RECA-1 used as an endothelial cell marker after CCH shows a short and irregular pattern in the brain. As shown in Figure 3B, western blot analysis showed that the expression of PECAM-1, PDGFRβ, occludin, claudin-9, ZO-1, and eNOS in the VaD group was significantly decreased in the hippocampus compared to the sham group. However, the Blockade of Lin28a increased the PECAM-1 (F = 57.746, *p* < 0.001), PDGFRβ (F = 18.319, *p* < 0.001), occludin (F = 43.067, *p* < 0.001), claudin-9 (F = 21.233, *p* < 0.001), ZO-1 (F = 79.594, *p* < 0.001), and eNOS (F = 10.52, *p* < 0.01) expression in the hippocampus. These results indicate that the blockade of Lin28a restored microvessels damage induced by BBB disruption.

### 3.4. Blockade of Lin28a Inhibits VaD-Induced Activation of Astrocyte and CCR6 in the Brain

Astrocyte and CCR6 expressions might be closely associated with neuroinflammatory processes in ischemic brain regions. As shown in Figure 4, the VaD group was significantly increased expression of GFAP (F = 25.911, *p* < 0.001) and CCR6 (F = 52.809, *p* < 0.001) compared to the sham group. The inhibition of Lin28a significantly reduced GFAP and CCR6 levels in the hippocampal dentate gyrus. These results suggest that VaD-induced inhibition of Lin28a overexpression moderated the activation of astrocytes and CCR6 in the brain.

### 3.5. Blockade of Lin28a Alleviates VaD-Induced Cognitive Impairment

At 42 days after BCCAO surgery, we performed passive avoidance and radial eight-armed maze tests to evaluate learning and memory in rats (Figure 5). The VaD group showed cognitive impairment in the passive avoidance and radial 8-arm maze tests compared to the sham group. As shown in Figure 5A, the latency time for rats staying in the light chamber decreased in the VaD group but increased in the Lin28a siRNA group (F = 19.539, *p* < 0.001). As shown in Figure 5B, the number of correct choices decreased and the number of errors increased in the VaD group. However, the Lin28a siRNA group increased the number of correct choices (F = 17.999, *p* < 0.001) and decreased the number of errors (F = 6.156, *p* < 0.01) compared to the VaD group. These results indicate that blockade of Lin28a attenuates VaD-induced cognitive impairment.

## 4. Discussion

In this study, overexpression of Lin28a was observed on days 4, 7, 14, and 28 after BCCAO. Blocking Lin28a ameliorates cognitive impairment, astrocyte activation, and BBB disruption induced by chronic cerebral hypoperfusion. Lin-28 is highly expressed during embryogenesis and early development in hypodermal, neural, and muscle cells, but gradually diminishes and disappears in adulthood [4]. However, Lin28a is overexpressed in the BCCAO-induced adult rat brain. Abnormally expressed Lin28a can stimulate tumor development and cause cell transformation [23]. Lin28a overexpression occurs with equal frequency in large fractions of tumors and inhibition of Lin28a and Lin28b via siRNA inhibits tumor growth [24].

Lin-28 level increases in the astrocyte after spinal cord injury and is involved in astrocyte activation and neuroinflammation [25]. Our results indicate that the expression levels of GFAP and Iba-1 increase along with the overexpression of Lin28a after chronic cerebral hypoperfusion. After intracerebral hemorrhage, Lin-28 is overexpressed and involved in inducing astrocyte proliferation [6]. Astrocyte activation plays a pivotal role in ischemic stroke injury [26]. The VaD group showed increased hypertrophic morphology of activated microglia and reactive astrocytes in the brain compared to the sham group. Blocking Lin28a significantly suppressed GFAP activation in the brain. Lin28a inhibition reduced the increase in astrocyte activity induced by chronic cerebral hypoperfusion. Our previous study confirmed the improvement in cognitive impairment by inhibiting VaD-induced oxidative stress, microglial activation, and reactive astrocytes [22]. This suppressed astrocyte hyperactivity and alleviated the cognitive impairment caused by VaD.

Chronic cerebral hypoperfusion causes energy depletion and neurodegeneration due to the overproduction of oxygen species and activated microglia, resulting in cognitive impairment [9]. Several studies have reported cognitive impairment through neuronal apoptosis induced by chronic cerebral hypoperfusion [27,28]. Apoptosis and autophagy are closely linked to long-lasting chronic cerebral hypoperfusion and are known to contribute to co-regulation of these events with microbalances [29,30,31]. However, in chronic cerebral hypoperfusion, autophagy activity has different roles depending on the time and amount [32]. The autophagy dysfunction occurred immediately after chronic cerebral hypoperfusion, and cognitive impairment appeared eight weeks after BCCAO [33]. The present study confirmed cognitive impairment using passive avoidance and radial 8-arm maze tests after chronic cerebral hypoperfusion. The passive avoidance test is a fear-aggravated test used to assess learning and memory in models of neurodegenerative diseases such as ischemic stroke and Alzheimer’s disease. The passive avoidance test 21 days after BCCAO surgery reported that the latency time to enter the dark chamber was significantly reduced in the BCCAO group compared to the sham group [21]. Furthermore, the latency time was significantly reduced in the VaD group on day 42 after BCCAO surgery. The radial 8-arm maze test evaluates working and reference memory. The test reported a cognitive decline 42 days after BCCAO surgery [22]. To confirm that BCCAO and siRNA treatment affected motor function, we used an open field test to show no difference in motor function in the sham, VaD, and VaD+Lin28a siRNA groups (Appendix A). We found that the blockade of Lin28a ameliorated the cognitive decline induced by BCCAO.

The endothelial cells forming the BBB in the central nervous system are highly specialized, precisely controlling substances entering and leaving the brain and effectively preventing harmful substances from entering the brain parenchyma [34]. Following ischemic stroke, tight junction protein complexes are disrupted, causing damage to the BBB, which increases the permeability of the brain microvascular endothelial cells [15]. BBB disruption is caused by the secretion of pro-inflammatory mediators, such as IL-1α, IL-1β, IL-6, and TNF-α, due to an increase in microglia and astrocytes in Alzheimer’s disease [35]. Tight junction proteins such as occludin, claudin, and ZO-1 play important roles in tight junction stability and functioning. BBB disruption results in the dissociation of ZO-1 by IL-1β [36]. Several studies have shown that the tight junction proteins alleviate cognitive impairment caused by chronic cerebral hypoperfusion by alleviating BBB damage [22,28]. In our study, the VaD group showed reduced expression levels of PECAM-1, PDGFRβ, occludin, claudin-9, and ZO-1, but the expression levels of junction proteins were restored when Lin28a overexpression was blocked. Lin28a overexpression induces the sustained translational inhibition of occludin, which regulates tight junctions, thereby disrupting the barrier function. The expression of occludin can be significantly inhibited by ectopic overexpression of Lin28a [37]. These results show that chronic cerebral hypoperfusion damages the BBB, but the blockade of Lin28a can ameliorate BBB disruption.

Hypoxia can impair BBB function by modulating expression levels of tight junction proteins, solute transporters, efflux transporters, and nutrient and hormone receptors [38]. Cells respond to hypoxia through hypoxia-inducing factor-1 (HIF-1), which consists primarily of the oxygen tension-regulating alpha subunit (HIF-1α). Under low oxygen conditions, HIF-1α stabilizes and migrates to the nucleus, regulating several genes, including those involved in angiogenesis, tumor proliferation, and apoptosis in cancer and ischemia [39]. A recent study reported that the induction of hypoxia increases the expression level of Lin28 protein [40]. Hypoxia stimulates neural stem cell proliferation by increasing HIF-1α expression, and the expression of Lin28 is regulated by HIF [41]. In cardiomyocytes, hypoxia induces the expression of lin28, inhibits miR-let-7, and impairs cell survival signals [42]. It has also been reported that inhibition of HIF-1 reduces BBB damage by regulating MMP-2 and VEGF [43]. In our study, we confirmed that inhibition of Lin28 reduced BBB disarrangement. The role of lin28 in BBB construction in the brain is still limited. However, our findings show potential for a novel lin28 function in neurodegeneration induced by hypoxic condition.

Following a brain injury, astrocytes are involved in brain inflammatory responses by stimulating microglia to upregulate the mRNA of CCR6 [18]. Activation of CCR6 induces BBB dysregulation and promotes the influx of multiple inflammatory cells by increasing the production of cytokines and chemokines [44]. CCR6 activation is associated majorly with early IL-17 production in acute infections and after ischemic stroke, IL-17 triggers the expression of TNF-α and chemokines. The inhibition of CCR6 reduces IL-17 production and ameliorates neurological damage. Neurologically beneficial effects can be obtained by inhibiting the increase of CCR6 levels after middle cerebral artery occlusion [20]. Our results showed that CCR6 overexpression in the brain was caused by VaD and CCR6 activity was reduced through Lin28a inhibition.

## 5. Conclusions

The results of these studies indicate that Lin28a overexpression after BCCAO was associated with cognitive impairment, astrocyte activation, and BBB disruption. However, neurological improvement was shown after inhibition of Lin28a overexpression. We suggest that the inhibition of overexpression of Lin28a induced by chronic cerebral hypoperfusion can alleviate VaD-related cognitive impairment by reducing the activation of astrocytes and CCR6 and by ameliorating BBB disruption.

## Figures and Tables

**Figure 1 biomedicines-10-00852-f001:**
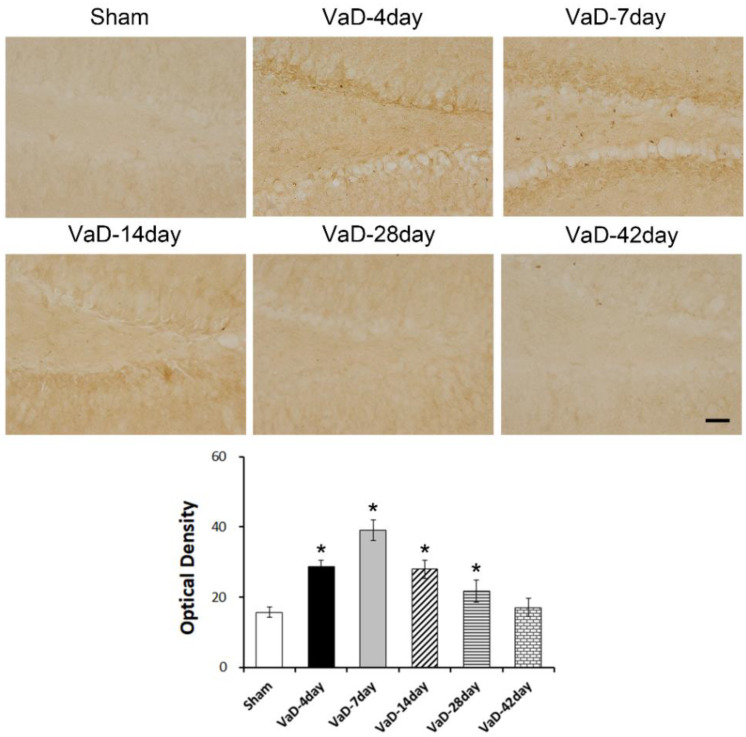
Expression of Lin28a in the hippocampal dentate gyrus after VaD. Representative images of Lin28a expression in the hippocampal dentate gyrus at 4, 7, 14, 28, and 42 days after BCCAO surgery or sham operation. Data are presented as mean ± S.E.M. * *p* < 0.05, compared to the sham group. A scale bar represents 50 μm.

**Figure 2 biomedicines-10-00852-f002:**
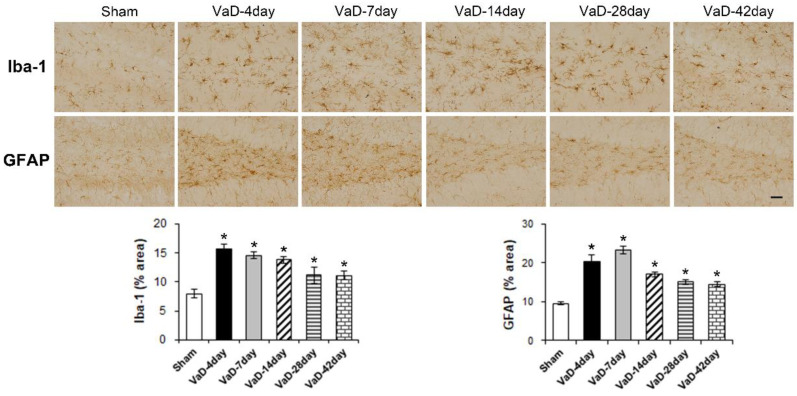
Expression of Iba-1 and GFAP in the hippocampal dentate gyrus after VaD. Representative images of Iba-1 and GFAP expression in the hippocampal dentate gyrus at 4, 7, 14, 28, and 42 days after BCCAO surgery or sham operation. Data are presented as mean ± S.E.M. * *p* < 0.05, compared to the sham group. A scale bar represents 50 μm.

**Figure 3 biomedicines-10-00852-f003:**
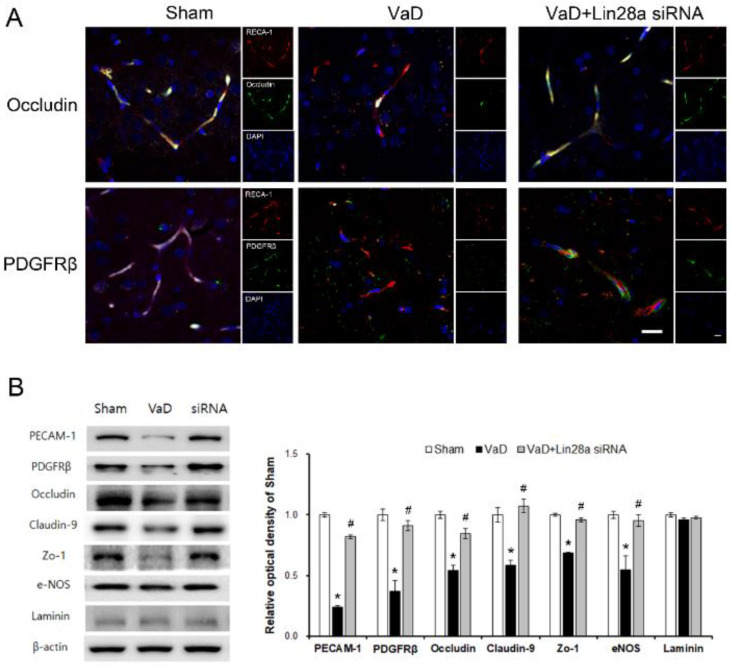
Blocking Lin28a ameliorates the loss of tight junction proteins in the brain. (**A**) Representative photographs of immunofluorescent staining of occludin (green), PDGFRβ (green), and RECA-1 (red) in the brain. (**B**) Representative bands of PECAM-1, PDGFRβ, occludin, claudin-9 and ZO-1, eNOS, and Laminin proteins in the brain were detected by western blotting. The data are presented as the mean ± S.E.M. * *p* < 0.05, compared to the sham group. # *p* < 0.05, compared to the VaD group. A scale bar represents 20 μm.

**Figure 4 biomedicines-10-00852-f004:**
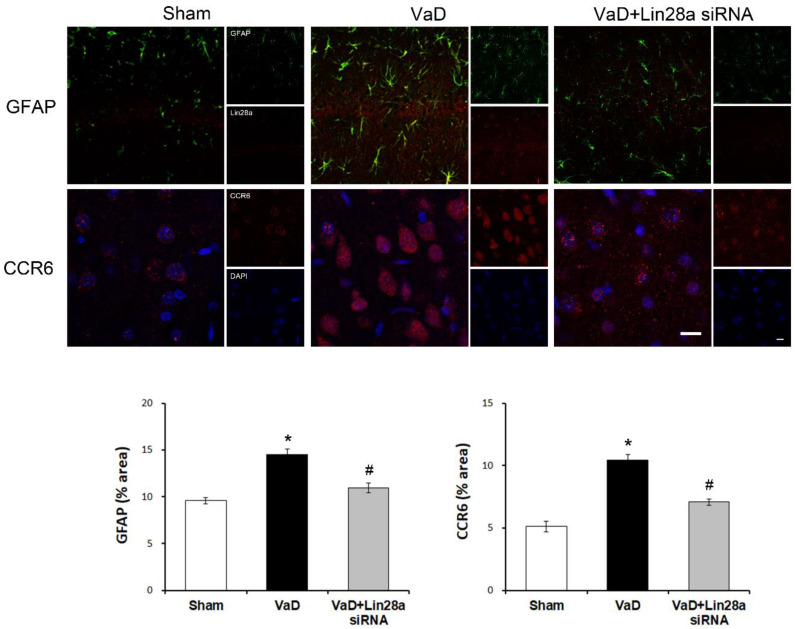
Blocking Lin28a inhibits the activation of GFAP and CCR6 in the brain after VaD. Representative photographs of immunofluorescent staining of GFAP (green), Lin28a (red), CCR6 (red), and DAPI (blue) in the brain. The data are presented as the mean ± S.E.M. * *p* < 0.05, compared to the sham group. # *p* < 0.05, compared to the VaD group. A scale bar represents 20 μm.

**Figure 5 biomedicines-10-00852-f005:**
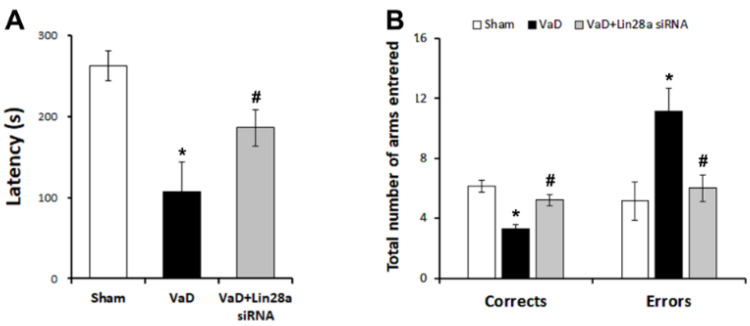
Effect of blocking Lin28a on VaD-induced cognitive impairment using passive avoidance and radial 8-arm maze tests. (**A**) In the passive avoidance test, VaD+Lin28a siRNA group show significantly improved latency times compared to the VaD group (**B**) In the radial 8-arm maze test, VaD+Lin28a siRNA group show significantly improved number of correct and errors choices compared to the VaD group. The data are presented as the mean ± S.E.M. * *p* < 0.05, compared to the sham group. # *p* < 0.05, compared to the VaD group.

## Data Availability

Not applicable.

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
