# Peer review of "Blockade on Lin28a Prevents Cognitive Impairment and Disruption of the Blood-Brain Barrier Induced by Chronic Cerebral Hypoperfusion"

_biomedicines, 2022, doi:10.3390/biomedicines10040852_

Round 1

Reviewer 1 Report

The purpose of this study has been to evaluate the effect on cognitive and brain functions by modulating Lin28a, which increases after chronic cerebral hypoperfusion.
The conclusions are interesting because they show how Lin28a siRNA treatment could inhibit VaD-induced microglial activity and alleviate blood-brain barrier damage. These results suggest that Lin28a blockade could be a new target to ameliorate cognitive impairment caused in vascular dementia.
I think it is interesting to be able to advance in one of the most prevalent pathologies and that cause more vital disability

Author Response

Response: We would like to thank reviewer for taking the time and effort necessary to review the manuscript. It is very important of your positive response to our study.

Reviewer 2 Report

The study by Lee et al. shows that that Lin28a overexpression after BCCAO is associated with cognitive impairment, astrocyte activation and BBB disruption. Inhibition of  Lin28a overexpression improvement neurological state of rats. Therefore, presented results are novel and may be valuable in the understanding of the role of Lin28a in the chronic cerebral hypoperfusion. The studies were performed on rats subjected to physiological tests of cognitive function, and collected biological material was used in western blot , IHC and IF studies. The results are clear and properly described, the conclusion is accurate. Nevertheless, I have some comments listed below:

1. The 2.4. section “Tissue preparation” should be rewritten as it contains numerous grammar mistakes.

2. IHC figures on Fig. 1 should be larger, or at higher magnification.

3. The fond size on bar figures should be increased.

4. The IF stainings on panel A of Figure 4 should be larger.

Author Response

  1. The 2.4. section “Tissue preparation” should be rewritten as it contains numerous grammar mistakes.

Response: We thank the reviewers for their careful review of the manuscript. According to reviewer’s advice, the 2.4 section "Tissue preparation" was changed as follows (page 3).

Etc. (Please check the manuscript, we highlighted the changes in red)

2.4. Tissue preparation

Rats were anesthetized by inhalation with ether, perfused with 0.01 M phosphate buffered saline (PBS), and the brains were removed. For western blot experiments, the samples were stored at -70°C until use. For immunohistochemistry, after perfusion with PBS, followed by additional perfusion with 4% paraformaldehyde (PFA), the brain was removed and fixed overnight in 4% PFA, followed by fixation in 30% sucrose solution. Subsequently, they were sliced into 40μm coronal sections on a cryostat (microtome, CM3050S, Leica, Nussloch, Germany).

  1. IHC figures on Fig. 1 should be larger, or at higher magnification.

Response: We made the IHC figures on Fig. 1 larger magnification in response to the comments of reviewer.

  1. The fond size on bar figures should be increased.

Response: Based on comments from reviewer, we increased the font size of the bar figures.

  1. The IF stainings on panel A of Figure 4 should be larger.

Response: We further enlarged the IF staining on panel A of Figure 4.

Reviewer 3 Report

The authors analysed Lin28a overexpression at different times of bilateral common carotid artery occlusion (BBCCAO). They found that blocking Lin28a could ameliorate cognitive impairment, astrocyte activation, and BBB disruption induced by chronic cerebral hypoperfusion. 

Interesting and well-presented data, although the novelty is not high. 

Tha main adresses are as follows: 

1) Which is the effect of Lin28 blockage on autophagic pathways in this experimental model?

2) the involvement of HIF1-aplha pathway should be addressed upon Lin28 siRNA

Author Response

1) Which is the effect of Lin28 blockage on autophagic pathways in this experimental model?

Response: First of all, we would like to thank Reviewer for taking the time and effort necessary to review the manuscript. In this study, we did not investigate the relevance of autophagy in neurodegeneration after 2VO. However, according to reviewer’s advice, the possibility of changes in autophagy due to BCCAO was applied to the discussion.

Etc. (Please check the manuscript, we highlighted the changes in red)

Discussion: page 8,

Apoptosis and autophagy are closely linked to long-lasting chronic cerebral hypoperfusion and are known to contribute to co-regulation of these events with microbalances [29–31]. However, in chronic cerebral hypoperfusion, autophagy activity has different roles depending on the time and amount [32]. The autophagy dysfunction occurred immediately after chronic cerebral hypoperfusion, and cognitive impairment appeared 8 weeks after BCCAO [33].

  1. Li, N.; Gu, Z.; Li, Y.; Fu, X.; Wang, J.; Bai, H. A modified bilateral carotid artery stenosis procedure to develop a chronic cerebral hypoperfusion rat model with an increased survival rate. J. Neurosci. Methods. 2015, 255, 115–121.
  2. Sui, X.; Kong, N.; Ye, L.; Han, W.; Zhou, J.; Zhang, Q.; He, C.; Pan, H. p38 and JNK MAPK pathways control the balance of apoptosis and autophagy in response to chemotherapeutic agents. Cancer Lett. 2014, 344, 174–179.
  3. Booth, L.A.; Tavallai, S.; Hamed, H.A.; Cruickshanks, N.; Dent, P. The role of cell signalling in the crosstalk between autophagy and apoptosis. Cell. Signal. 2014, 26, 549–555.
  4. Ferrucci, M.; Biagioni, F.; Ryskalin, L.; Limanaqi, F.; Gambardella, S.; Frati, A.; Fornai, F. Ambiguous Effects of Autophagy Activation Following Hypoperfusion/Ischemia. Int. J. Mol. Sci. 2018, 19, 2756. https://doi.org/10.3390/ijms19092756.
  5. Zou, W.; Song, Y.; Li, Y. et al. The Role of Autophagy in the Correlation Between Neuron Damage and Cognitive Impairment in Rat Chronic Cerebral Hypoperfusion. Mol Neurobiol. 2018, 55, 776–791. https://doi.org/10.1007/s12035-016-0351-z.

2) the involvement of HIF1-aplha pathway should be addressed upon Lin28 siRNA

Response: In response to the comments of the reviewers, we have added information related to the HIF1-α pathway and Lin28 in the discussion section. Importance of HIF1-α in the ischemia model, we had to confirmed that relationship between HIF1-α and Lin28. A recent study reported that the induction of hypoxia increases the expression level of Lin28 protein (Lin and Chen, 2020). Hypoxia stimulates neural stem cell proliferation by increasing HIF-1α expression, and the expression of Lin28 is regulated by HIF (Qi et al., 2017). It would be a better study if the factors related to the HIF1-α pathway are identified in the future studies of chronic cerebral hypoperfusion.

Etc. (Please check the manuscript, we highlighted the changes in red)

Discussion: page 9,

Hypoxia can impair BBB function by modulating expression levels of tight junction proteins, solute transporters, efflux transporters, and nutrient and hormone receptors [38]. Cells respond to hypoxia through hypoxia-inducing factor-1 (HIF-1), which consists primarily of the oxygen tension-regulating alpha subunit (HIF-1α). Under low oxygen conditions, HIF-1α stabilizes and migrates to the nucleus, regulating several genes, including those involved in angiogenesis, tumor proliferation, and apoptosis in cancer and ischemia [39]. A recent study reported that the induction of hypoxia increases the expression level of Lin28 protein [40]. Hypoxia stimulates neural stem cell proliferation by increasing HIF-1α expression, and the expression of Lin28 is regulated by HIF [41]. In cardiomyocytes, hypoxia induces the expression of lin28, inhibits miR-let-7, and impairs cell survival signals [42]. It has also been reported that inhibition of HIF-1 reduces BBB damage by regulating MMP-2 and VEGF [43]. In our study, we confirmed that inhibition of Lin28 reduced BBB damage. The role of lin28 in BBB disruptions in the brain is still limited. However, our findings show potential for a novel lin28 function in neurodegeneration induced by hypoxic condition.

  1. Engelhardt, S.; Al-Ahmad, A.J.; Gassmann, M.; Ogunshola, O.O. Hypoxia selectively disrupts brain microvascular endothelial tight junction complexes through a hypoxia-inducible factor-1 (HIF-1) dependent mechanism. J. Cell. Physiol. 2014, 229, 1096–1105, doi:10.1002/jcp.24544.
  2. Ziello, J.E.; Jovin, I.S.; Huang, Y. Hypoxia-Inducible Factor (HIF)-1 regulatory pathway and its potential for therapeutic intervention in malignancy and ischemia. Yale J. Biol. Med. 2007, 80, 51–60.
  3. Lin, C.Q.; Chen, L.K. Cerebral dopamine neurotrophic factor promotes the proliferation and differentiation of neural stem cells in hypoxic environments. Neural Regen. Res. 2020, 15, 2057–2062, doi:10.4103/1673-5374.282262.
  4. Qi, C.; Zhang, J.; Chen, X.; Wan, J.; Wang, J.; Zhang, P.; Liu, Y. Hypoxia stimulates neural stem cell proliferation by increasing HIF-1α expression and activating Wnt/β-catenin signaling. Cell. Mol. Biol. 2017, 63, 12–19, doi:10.14715/cmb/2017.63.7.2.
  5. Hyochol Ahn, PhD, Michael Weaver, PhD, Debra Lyon, PhD, Eunyoung Choi, RN, and Roger B. Fillingim, P.; Tumbar, C.J.F.S.T. A cardiac myocyte-restricted Lin28/let7 regulatory axis promotes hypoxia-mediated apoptosis by inducing the AKT signaling suppressor PIK3IP1. Physiol. Behav. 2017, 176, 139–148, doi:10.1016/j.bbadis.2015.12.004.A.
  6. Shen, Y.; Gu, J.; Liu, Z.; Xu, C.; Qian, S.; Zhang, X.; Zhou, B.; Guan, Q.; Sun, Y.; Wang, Y.; et al. Inhibition of HIF-1α reduced blood brain barrier damage by regulating MMP-2 and VEGF during acute cerebral ischemia. Front. Cell. Neurosci. 2018, 12, 1–8, doi:10.3389/fncel.2018.00288.

Round 2

Reviewer 2 Report

I accept the manuscript in the present form.

Author Response

Thank you for accepting our manuscript.

Reviewer 3 Report

Vascular dementia has been associated to oxidative stress and misfolded protein accumulation (in particular, amyloid beta). Which is the effect of n Lin28a on these pathways? these issue should be addressed. 

Author Response

Response: Vascular dementia (VaD) is caused by chronic cerebral hypoperfusion, leading to multiple microinfarcts and vascular cognitive impairment (Kalaria et al., 2016; Van et al., 2018). Vascular dementia can sometimes be triggered by cerebral amyloid angiopathy. Cerebral amyloid angiopathy is a cerebrovascular disorder caused by the accumulation of cerebral amyloid beta in the cortical vessels of the brain. Cerebral ischemia results in protein misfolding protein accumulation, aggregation, and impairment of autophagy (Luo et al., 2013). Impairment of autophagy contributes to abnormal protein aggregation and organelle damages after brain ischemia (Liu et al., 2010). Misfolded protein accumulation and aggregation are hallmarks of many neurodegenerative diseases (Dohm et al., 2008). Several studies have reported increased amyloid deposition in stroke patients, implicating that ischemia promotes AD pathology (Kalaria et al., 2000; Aho et al., 2006). Although we did not see amyloid beta in our study, we could not found that previous study of a direct relationship between amyloid beta and Lin28a but BBB damage is driven by accumulation of amyloid beta (Blair et al., 2015). In our results, BBB destruction was observed with Lin28a overexpression increased after VaD, and it was confirmed that BBB damage was improved after suppressing Lin28a overexpression. Autophagy is the main degradation pathway responsible for eliminating abnormal protein aggregates and damaged organelles prevalent in neurons after cerebral ischemia (Liu et al., 2010). Lin28 suppresses autophagy to promote cellular growth (Jun-Hao et al., 2016). Lin28 blocks let-7 expression, while let-7 itself binds to the 3′ UTR of Lin28 mRNA to regulate negatively Lin28 expression. Let-7 activates autophagy by coordinately down-regulating the amino acid sensing pathway to prevent mTORC1 activation. Let-7 induced autophagy in the brain to eliminate protein aggregates, establishing its physiological relevance for in vivo autophagy modulation (Dubinsky et al., 2014). Therefore, it would be good to conduct research on the correlation between VaD-induced accumulation of abnormal protein accumulation and overexpression of Lin28a in the future by referring to the opinions of reviewer.

Round 3

Reviewer 3 Report

the manuscirpt has been improved and it is now suitable for publication